# Maritime Paleoindian technology, subsistence, and ecology at an ~11,700 year old Paleocoastal site on California's Northern Channel Islands, USA

**Jon M. Erlandson**[1,2], **Todd J. Braje**[3], **Amira F. Ainis**[1,2], **Brendan J. Culleton**[4], **Kristina M. Gill**[1,5], **Courtney A. Hofman**[6,7], **Douglas J. Kennett**[8], **Leslie A. Reeder-Myers**[9], **Torben C. Rick**[10]*

1 Museum of Natural and Cultural History, University of Oregon, Eugene, OR, United States of America, 2 Department of Anthropology, University of Oregon, Eugene, OR, United States of America, 3 Department of Anthropology, San Diego State University, San Diego, CA, United States of America, 4 Institutes of Energy and the Environment, Pennsylvania State University, University Park, PA, United States of America, 5 Santa Barbara Botanic Garden, Santa Barbara, CA, United States of America, 6 Department of Anthropology, University of Oklahoma, Norman, OK, United States of America, 7 Laboratories of Molecular Anthropology and Microbiome Research, University of Oklahoma, Norman, OK, United States of America, 8 Department of Anthropology, University of California, Santa Barbara, CA, United States of America, 9 Department of Anthropology, Temple University, Philadelphia, PA, United States of America, 10 Department of Anthropology, National Museum of Natural History, Smithsonian Institution, Washington, DC, United States of America

* rickt@si.edu

## Abstract

During the last 10 years, we have learned a great deal about the potential for a coastal peopling of the Americas and the importance of marine resources in early economies. Despite research at a growing number of terminal Pleistocene archaeological sites on the Pacific Coast of the Americas, however, important questions remain about the lifeways of early Paleocoastal peoples. Research at CA-SRI-26, a roughly 11,700 year old site on California's Santa Rosa Island, provides new data on Paleoindian technologies, subsistence strategies, and seasonality in an insular maritime setting. Buried beneath approximately two meters of alluvium, much of the site has been lost to erosion, but its remnants have produced chipped stone artifacts (crescents and Channel Island Amol and Channel Island Barbed points) diagnostic of early island Paleocoastal components. The bones of waterfowl and seabirds, fish, and marine mammals, along with small amounts of shellfish document a diverse subsistence strategy. These data support a relatively brief occupation during the wetter "winter" season (late fall to early spring), in an upland location several km from the open coast. When placed in the context of other Paleocoastal sites on the Channel Islands, CA-SRI-26 demonstrates diverse maritime subsistence strategies and a mix of seasonal and more sustained year-round island occupations. Our results add to knowledge about a distinctive island Paleocoastal culture that appears to be related to Western Stemmed Tradition sites widely scattered across western North America.

## Introduction

Many archaeologists now believe that the initial peopling of the Americas followed a coastal route around the Pacific Rim from Northeast Asia into the Americas [1]. A growing body of

**Data Availability Statement:** All data are contained within the text and tables of this manuscript. The collections are currently housed at the University of

Oregon and will be permanently curated at the Santa Barbara Museum of Natural History.

**Funding:** This research was supported by a grant from the National Science Foundation (BCS-0917677) awarded to JME and TCR, the US National Park Service (Cooperative Agreement # H812005033 to JME), and the Paleoindian Research Endowment at the University of Oregon's (UO) Museum of Natural & Cultural History (to JME). The funders had no role in study design, data collection and analysis, decision to publish, or preparation of the manuscript.

**Competing interests:** The authors have declared that no competing interests exist.

archaeological, geological, and genetic data support the feasibility of such a coastal dispersal, as does extensive evidence for seafaring by anatomically modern humans (AMH) from islands of East Asia and greater Australia between ~55,000 and 30,000 years ago [2]. With global sea level rising ~100–120 m in the last 20,000 years, identifying Late Pleistocene archaeological sites in coastal settings is difficult, but recent research demonstrates that several parts of North America's Pacific Coast were occupied by humans between ~14,000 and 13,000 years ago [3]. Monte Verde II in Chile, a peri-coastal site occupied ~14,500 to 14,000 years ago, has produced several types of edible seaweed and supports a coastal dispersal [4]. So far, however, the earliest Pacific Coast sites have produced only tentative evidence for technological connections to coastal Northeast Asia, although some have proposed a potential connection between terminal Pleistocene Pacific Rim assemblages containing stemmed (or tanged) points, crescents (lunates), and leaf-shaped (foliate) bifaces [3, 5, 6].

In the last two decades, coastal southern California has produced a variety of archaeological data supporting a Late Pleistocene human occupation in the New World, including evidence for early maritime economies and seafaring. The Arlington Springs site (CA-SRI-173) produced three human bones from a single individual dated to nearly 13,000 cal BP [7], but no diagnostic technology. Erlandson et al. [8] identified a maritime Paleoindian tradition on California's Northern Channel Islands (NCI) that contains stemmed points, crescents, and leaf-shaped bifaces dating as early as 12,200 cal BP, artifacts that appear to be linked to the Western Stemmed Tradition (WST) of western North America. More than 100 Paleocoastal sites (~13,000–8,000 cal BP) have been documented on the NCI, but only a few sites older than 11,000 cal BP have produced sizable, well-dated artifact and faunal assemblages [9]. One of these, CA-SRI-512, a buried site near the mouth of Arlington Canyon dated to ~11,700 cal BP, is estimated to have been located as much as 5–6 km from the ancient coast, and has produced scores of bifaces (CIBs, crescents, and foliates) and thousands of bird bones. CA-SRI-512 remains something of an anomaly among early island Paleocoastal sites, however, leading some to question the commitment to seafaring and maritime subsistence among early islanders (e.g., [10, 11]).

Here, we report on a buried ~11,700 year old Paleocoastal component identified at CA-SRI-26, also located near the mouth of Arlington Canyon, that has produced artifacts and faunal remains similar to the assemblage at CA-SRI-512. We describe the discovery, dating, and contents of the Paleocoastal component at CA-SRI-26, then discuss its relevance for understanding early coastal adaptations in the Americas.

## Site setting, stratigraphy, and dating

Today the NCI—consisting of Anacapa, Santa Cruz, Santa Rosa, and San Miguel islands—are located between 20 and 44 km from the adjacent mainland coast (Fig 1). Near the end of the Last Glacial Maximum (ca 20,000 cal BP), however, the islands were united into a single ~125 km long island known as Santarosae [12], the eastern end of which was as little as 6–8 km from the mainland. The latest reconstructions of Santarosae's paleogeography suggest that rising post-glacial sea level reduced the landmass by as much as 70–75 percent, causing the NCI to separate between about 11,000 and 9,000 years ago [13, 14]. Then and now, the NCI had a relatively impoverished terrestrial fauna, a fairly diverse and productive flora, and a wealth of edible marine resources, from seaweeds to marine mammals, shellfish, fish, and seabirds [15, 16].

CA-SRI-26 is currently located approximately 60 m (~200 ft) above sea level on a bluff overlooking the ocean, roughly 200 m east of Arlington Canyon where a rocky sill in the canyon bottom brings freshwater to the surface year-round. A high sea cliff, broad sand beach, and the

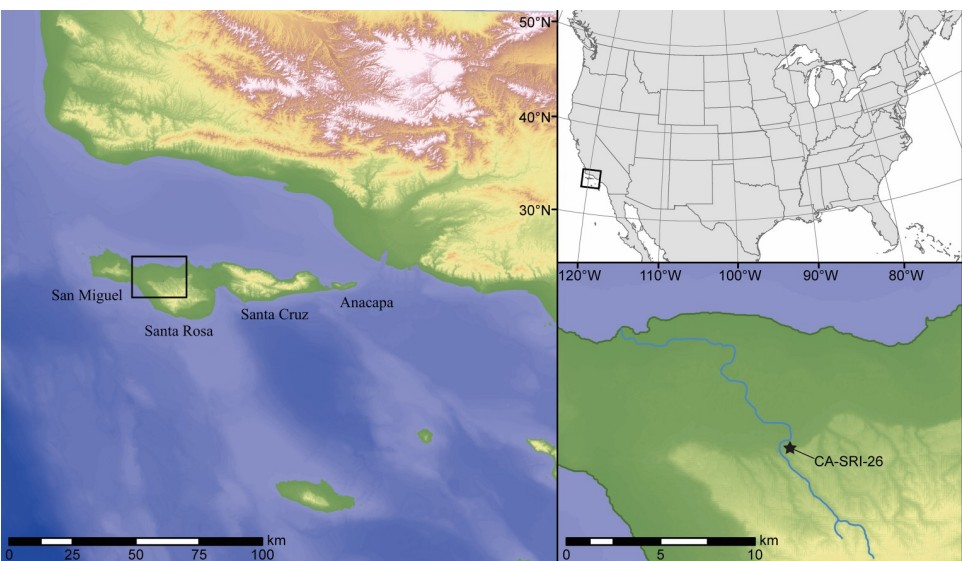

**Fig 1. Estimated position of Alta California's Channel Islands and mainland coast when CA-SRI-26 was occupied, and the approximate course of Arlington Creek.** The paleo-digital elevation model, which forms the image background and is the basis for both the shoreline and stream estimates, was produced using the methods from Reeder-Myers et al. [13], but with an updated model of glacio-isostatic adjusted sea level from Argus et al. [17] and Peltier et al. [18]. CA-SRI-26 was about 6 km from the contemporary shoreline in a straight line, but about 15.5 km if following the course of the stream to its mouth. Relative sea level ~11,700 years ago on the northwest coast of Santa Rosa Island is estimated at -60 ± 5 m and highlighted in dark green. Drafted by Leslie Reeder-Myers.

coastline are located just north of the site. At about ~11,700 years ago, however, sea level around the NCI was ~60 m below present [13] and the site may have been as much as 4–6 km from the outer coast. Rapidly rising postglacial seas may have formed an embayment at the mouth of Arlington Canyon, however, backed by a broad, marshy canyon bottom that was attractive to Paleocoastal peoples for its access to freshwater and a variety of marine, wetland, and terrestrial resources [8].

CA-SRI-26 was first recorded in the 1940s by Phil Orr of the Santa Barbara Museum of Natural History. Although he never excavated or dated the site, Orr [12, 19] described CA-SRI-26 as potentially the oldest of several 'Pleistocene' shell middens he documented on the northwest coast of Santa Rosa Island. Erlandson [19] revisited these midden sites in the early 1990s, collected radiocarbon (¹⁴C) samples, and demonstrated that all contained Early Holocene components dated between ~9300 and 7700 cal BP. The youngest of these was a patchy red abalone shell midden at CA-SRI-26 found in a buried paleosol ~1–1.5 m below the surface [20, 21]. In deeply incised gullies nearby, multiple paleosols were visible deeper in Late Pleistocene alluvial sediments of the Upper Tecolote Formation, but no archaeological materials were found in these older soils at the time (Fig 2).

In 2010, Erlandson re-examined several eroding gullies at CA-SRI-26 and identified a low-density deposit of chipped stone artifacts, animal bones, and occasional marine shells eroding from a ~40–50 cm thick A4 paleosol situated ~2–2.5 m below the surface, roughly a meter below the Early Holocene red abalone midden. This Paleocoastal component produced a relatively small but significant assemblage of artifacts and faunal remains from stratified contexts. These included a weathered California mussel (*Mytilus californianus*) shell fragment ¹⁴C dated to 10,545 ± 30 RYBP (UCIAMS-80937), with a calibrated age range of 11,420–11,155 cal BP (Table 1). A second date of 10,150 ± 70 RYBP (OS-96885) for purified collagen extracted from a goose (*Branta* spp.) bone from the Paleocoastal component was several centuries older, with

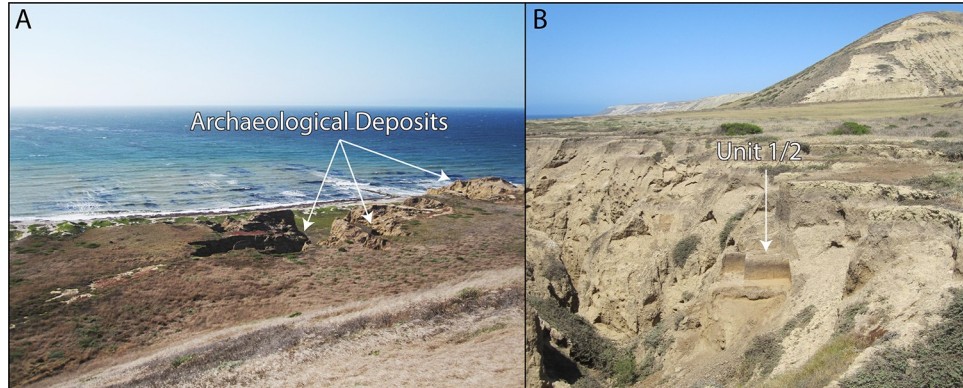

**Fig 2.** A) The modern setting of CA-SRI-26, with sea cliff, shoreline, and Santa Barbara Channel in the background, alluvium covered marine terrace cut by gullies, and terrace riser (ancient sea cliff) rising in the foreground (photo by J. Erlandson). B) Stratigraphy in gully wall at CA-SRI-26, showing location of contiguous Test Units 1 (center right) and 2 (center left) in A4 soil horizon, built in thick Late Pleistocene and Holocene alluvium deposited on a raised marine terrace, with a terrace riser (ancient sea cliff) above CA-SRI-512 in the distance (photo by J. Erlandson).

a $^{13}C/^{12}C$ ratio (-22.55‰) typical of terrestrial food webs, and a 2-sigma calibrated calendar age range of 11,980–11,640 cal BP. Because the dated mussel shell still showed signs of weathering even after heavy etching in hydrochloric acid, we hypothesized that the older bone date most accurately reflected the age of the Paleocoastal component at CA-SRI-26. We submitted a second marine shell sample, composed of a single well-preserved fragment of red abalone (*Haliotis rufescens*) shell epidermis, for AMS $^{14}C$ dating. Analysis of this sample provided a date of 10,700 ± 37 RYBP (D-AMS-8725), with a calibrated age range of 11,805 to 11,305 cal BP, but a 93 percent probability that the sample is >11,700 years old. Finally, we obtained a second date on purified collagen from another goose bone of 10,070 ± 30 RYBP (UCIAMS-94043), with a calibrated age range of 11,755–11,415 cal BP and an 80 percent probability that the sample was >11,600 years old. Together, the two bone dates and the second shell date all overlap earlier in the wider calendar age range, suggesting that CA-SRI-26 was most likely occupied roughly 11,700 years ago, give or take a century. The contents and stratigraphy of this component are similar to a slightly older Paleocoastal assemblage from CA-SRI-512

**Table 1.** $^{14}C$ dates from terminal Pleistocene and Early Holocene components at CA-SRI-26.

| Provenience | Material Dated | Lab No. | Measured Age | Calendar Age (cal BP, 2sigma) |
|---|---|---|---|---|
| Locus B, A3 soil | Shell: *Haliotis rufescens* | Beta-47819 | 7620 ± 80 | 7975–7650 |
| Locus B, A3 soil | Shell: *Haliotis rufescens* | OS-96946 | 7760 ± 40 | 8065–7845 |
| Locus C, A3 soil | Shell: *Mytilus californianus* | OS-96947 | 9540 ± 40 | 10,210–9935 |
| Locus A, A4 soil | Shell: *Mytilus californianus* | UCIAMS-80937 | 10,545 ± 30* | 11, 420–11,155 |
| Locus A, A4 soil | Shell: *Haliotis rufescens* | D-AMS-8725 | 10,700 ± 37 | 11,805–11,305 |
| Locus A, A4 soil | Purified bone collagen: Goose | UCIAMS-94043 | 10,070 ± 30¶ | 11,755–11,415 |
| Locus A, A4 soil | Purified bone collagen: Goose | OS-96885 | 10,150 ± 70¶ | 11,980–11,640 |

all dates are for single shell or bone fragments, measured via accelerator mass spectrometry; calendar age ranges were calculated using the marine calibration curve for shell samples and terrestrial northern hemisphere curve for bone samples in CALIB 6.0 [22, 23], with a ΔR of 261 ± 21 [24] for marine samples

*suspected to be too young due to poor shell preservation

¶C/N = 3.34 for UCIAMS-94043; UCIAMS-94043 processed by XAD and OS-96885 processed by EDTA.

located ~400 m to the east, supporting the veracity of the $^{14}$C chronology for the basal component at CA-SRI-26.

## Field and laboratory methods

Our research was conducted under Archaeological Resources Protection Act (ARPA) Permit PWR-1979-09-CA-04 to JME and TCR. Because the portion of CA-SRI-26 that we excavated has been—and continues to be—heavily affected by erosion in a deep gully system, we conducted limited excavations to understand the nature of the threatened deposits. This work consisted of clearing a stratigraphic profile approximately two meters wide in a nearly vertical section of the eroding gully wall and the excavation of two test units and four bulk samples. The bulk samples consisted of 20 liters of soil from each of four 10 cm levels. Later, three contiguous ~50 x100 cm test units (Units 1A, 1B, and 2) were excavated adjacent to the bulk sample location. Although the excavated sample from the site is small (~0.83 m$^3$), it is supplemented by collections of artifacts and faunal remains from eroding gully walls.

Faunal remains were collected from the early Paleocoastal component at CA-SRI-26 using three primary strategies: recovery from near-vertical surfaces of the A4 soil horizon exposed in eroding stratigraphic profiles in gully walls within the site area (Surface); bulk soil samples collected from these eroding stratigraphic exposures; and the three test units excavated through the A4 soil (see Fig 2). The excavated soils were all dry-screened over 1/16-inch mesh in the field, with screen residuals returned to the University of Oregon for detailed analysis.

Excavations, along with study of extensive gully wall exposures within the site boundaries, suggest that the relatively low-density Paleocoastal component at CA-SRI-26 probably represents a single occupation. The low density of cultural materials is partly a function of site formation processes. The Channel Islands have a dearth of burrowing animals and the unusual stratigraphic integrity of many of their archaeological sites [25]. At CA-SRI-26, there is little or no evidence for stratigraphic mixing between discrete soil horizons, but within the A4 soil both artifacts and faunal remains are found scattered throughout the 40–50 cm thick soil, atypical of many Channel Island soils where dense but thin midden layers are common. This clay-rich A4 soil appears to have been churned by argilliturbation, caused by the seasonal shrinking and swelling of the soil in a Mediterranean climate characterized by wet winters and dry summers [26]. Argilliturbation may also be responsible for the relatively heavy fragmentation of bird bones and the near absence of charcoal in the soil, despite the fact that some of the recovered bone fragments were burned. All of our research took place in an area of the site away from overlying shell middens in the A3 soil, limiting the possibility that any of these materials came from the ~10,000 or ~8000 year old site components.

## Paleocoastal artifacts and technology

The artifacts from CA-SRI-26 document maritime Paleoindian technologies on the NCI. Neither Arlington Springs (CA-SRI-173) nor a Terminal Pleistocene component at Daisy Cave (CA-SMI-261) produced diagnostic artifacts, but recent work at the nearby CA-SRI-512 and the Cardwell Bluffs site complex (CA-SMI-678, -679, and -701) on eastern San Miguel Island has shown that Paleocoastal islanders were equipped with finely made stemmed points and chipped stone crescents generally similar to those found in WST sites of western North America [5, 27, 28].

The CA-SRI-26 assemblage is smaller, but similar in many respects to the assemblages from CA-SRI-512 and, to an extent, Cardwell Bluffs. An extraordinary, ultra-thin stemmed and serrated Channel Island Amol (CIA) point (Fig 3) found on an eroded surface immediately north of the intact site remnants is one of only ~25 CIA points currently known from the NCI [29].

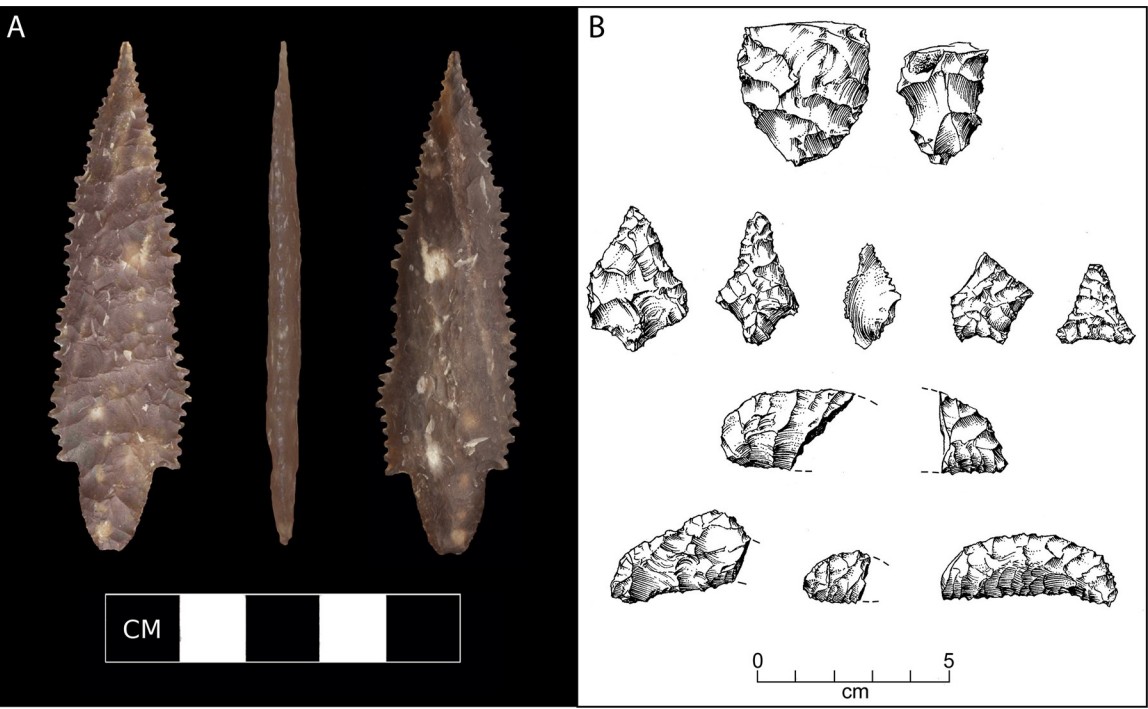

**Fig 3.** A) A Channel Island Amol (CIA) point found on an eroded gully surface just north of the buried Paleocoastal midden area at CA-SRI-26 (digital photo composite by K. Hamm & J. Erlandson). B) Bifaces from the ~11,700 cal BP component at CA-SRI-26: biface preforms (top), CIB preforms or fragments (middle); and crescents or crescentic artifacts (bottom rows). Drawn by R. Van Rossman.

In the only two cases where CIA points have been found in situ, they come from shell midden deposits dated to ~12,000 cal BP (Erlandson, unpublished notes). Several fragments or pre-forms of Channel Island Barbed (CIB) points have also been found at the site, along with three chipped stone crescent fragments—two from stratified contexts and another from the gully wall. Chipped stone crescents are a unique and relatively rare artifact found in early sites of the western United States, from Baja and Alta California to the Great Basin and Pacific Northwest [30]. Crescents are closely associated with wetland habitats [31] and on the NCI they probably were used primarily as transverse projectile points for hunting waterfowl or seabirds [5]. The CIA and CIB points could also have been used for bird hunting, but more likely as dart tips for fishing or the hunting of sea otters and pinnipeds [32].

## Subsistence and ecology

Our excavations produced a total of 715 animal bones (or fragments) weighing 91.59 g, along with 82.11 g of marine shell (Table 2). The shell sample, too small to be considered representative, contained just two shellfish species, red abalone (*Haliotis rufescens*) and California mussel (*Mytilus californianus*), along with minute amounts of unidentified shell. Both these species are among the top-ranked shellfish available to humans on the NCI [33]. In this small assemblage, nearly 75 percent of the marine shell comes from red abalone, a low intertidal or subtidal species closely associated with rocky shores and kelp forests.

The vertebrate assemblage contains considerably greater diversity than the shellfish assemblage, including the remains of birds, fish, marine mammal, small rodents, and very small amounts ($n = 6$; 0.39 g) of undifferentiated bone. The most abundant of these general taxonomic categories is bird bone (NISP = 353, 62.29 g), found in every sample, but the bones

**Table 2. Faunal remains from the ~11,700 year old Paleocoastal component at CA-SRI-26 (weights in grams).**

| | Surface | | Bulk Sample | | Unit 1 | | Unit 1a | | Unit 2 | | Total | |
|---|---|---|---|---|---|---|---|---|---|---|---|---|
| Faunal Category | NISP | Wt | NISP | Wt | NISP | Wt | NISP | Wt | NISP | Wt | NISP | Wt |
| Marine mammal (undiff.) | 1 | 2.12 | 5 | 1.07 | 10 | 2.13 | 1 | 0.14 | 3 | 2.85 | 20 | 8.31 |
| Mouse (*Peromyscus nesodytes*) | 4 | 0.29 | 21 | 0.9 | 3 | 0.07 | 10 | 0.36 | 15 | 0.64 | 53 | 2.26 |
| Small mammal (rodent, undiff.) | 0 | 0 | 4 | 0.08 | 4 | 0.14 | 0 | 0 | 0 | 0 | 8 | 0.22 |
| **Mammal bone subtotal** | **5** | **2.41** | **30** | **2.05** | **17** | **2.34** | **11** | **0.5** | **18** | **3.49** | **81** | **10.79** |
| Goose (*Branta* spp.) | 7 | 5.3 | 0 | 0 | 1 | 2.83 | 0 | 0 | 0 | 0 | 8 | 8.13 |
| Bird (Aves, undiff.) | 40 | 18.79 | 109 | 11.44 | 84 | 13.09 | 6 | 0.71 | 106 | 10.13 | 345 | 54.16 |
| **Bird bone subtotal** | **47** | **24.09** | **109** | **11.44** | **85** | **15.92** | **6** | **0.71** | **106** | **10.13** | **353** | **62.29** |
| Rockfish (*Sebastes* spp.) | 1 | 0.73 | 10 | 2.02 | 10 | 1.71 | 3 | 0.64 | 20 | 2.93 | 44 | 8.03 |
| Greenling (*Hexagrammos* spp.) | 0 | 0 | 0 | 0 | 0 | 0 | 0 | 0 | 2 | 0.1 | 2 | 0.1 |
| Cabezon? (*Scorpaenichthys marmoratus*) | 0 | 0 | 0 | 0 | 1 | 0.21 | 0 | 0 | 0 | 0 | 1 | 0.21 |
| Pile perch (*Rhacochilus vacca*) | 0 | 0 | 0 | 0 | 0 | 0 | 0 | 0 | 1 | 0.02 | 1 | 0.02 |
| Fish (undiff.) | 3 | 0.23 | 47 | 1.87 | 47 | 2.53 | 19 | 0.92 | 112 | 4.42 | 228 | 9.97 |
| **Fish subtotal** | **4** | **0.96** | **57** | **3.89** | **58** | **4.45** | **22** | **1.56** | **135** | **7.47** | **276** | **18.33** |
| Bone undiff. | 0 | 0 | 2 | 0.01 | 1 | 0.1 | 2 | 0.07 | 0 | 0 | 5 | 0.18 |
| **Bone total** | **56** | **27.46** | **198** | **17.39** | **161** | **22.81** | **41** | **2.84** | **259** | **21.09** | **715** | **91.59** |
| Red abalone (*Haliotis rufescens*) | | 2.9 | | 0 | | 2.84 | | 50.43 | | 4.84 | | 61.01 |
| California mussel (*Mytilus californianus*) | | 5.6 | | 5.6 | | 2.51 | | 0.07 | | 7.27 | | 21.05 |
| Marine shell, undiff. | | 0 | | 0 | | 0 | | 0 | | 0.05 | | 0.05 |
| **Shell total** | | **8.5** | | **5.6** | | **5.35** | | **50.5** | | **12.16** | | **82.11** |

were generally heavily fragmented. Only eight bird bones could be identified to a specific taxon, all from goose (*Branta* spp.). Most of the undifferentiated bird bone fragments are of comparable size and probably represent waterfowl or relatively large seabirds hunted by the site occupants. As is the case at CA-SRI-512, the relative abundance of geese and waterfowl, which are migratory and visit the NCI from late fall to early spring, suggests that CA-SRI-26 may have been occupied during the relatively wetter and cooler "winter" season.

Among the fish remains, 276 specimens weighing 18.34 g were recovered. Most of these were undifferentiated teleost remains, but the identifiable bones come from four taxa: rockfish (*Sebastes* spp., 96% of identifiable fish by weight), greenling (*Hexagrammos* ssp., 1.2%), pile perch (*Rhacochilus vacca*, <1%) and what is probably cabezon (*Scorpaenichthys marmoratus*; 2.5%). These fish, especially the rockfish, can be caught in a variety of marine habitats, but they are most likely to have come from nearshore kelp forests and rocky reefs.

Marine mammal bone fragments were also found in every excavated context, with 20 fragments weighing 8.31 g. The marine mammal remains are all too fragmented to be identified to even broad taxonomic categories (e.g., pinnipeds vs. cetaceans) with traditional zooarchaeological methods, but most are probably from one or more of the six seal or sea lion species present today in island waters. Two of the larger and seemingly best-preserved bone fragments were analyzed with Zooarchaeology Mass Spectrometry (ZooMS) to determine species [34]. One of these had poor collagen preservation and could not be identified, but the second was identified as elephant seal (*Mirounga angustirostris*) [35]. The identification of elephant seal in this early Paleocoastal component is significant as they are rare in later Pacific Coast archaeological sites, but abundant and highly vulnerable on the NCI today [32, 35, 36].

Other mammal remains were limited to 61 small mammal bones, all consistent with small rodent bones. The small rodent remains are ubiquitous and are almost certainly natural site constituents, although they may also have been attracted to the site by the rewards of human

**Table 3. Estimated edible meat yields for faunal remains recovered from the ~11,700 year old Paleocoastal component at CA-SRI-26.**

| Faunal Category | Weight (g.) | Multiplier | Meat Weight (g.) | %Meat Weight |
|---|---|---|---|---|
| Bird | 62.29 | 15 | 934.4 | 53.9 |
| Fish | 18.33 | 27.7 | 507.7 | 29.3 |
| Marine mammal | 8.31 | 24.2 | 201.1 | 11.6 |
| Red abalone | 61.01 | 1.36 | 83.0 | 4.8 |
| Mussel | 21.05 | 0.298 | 6.3 | 0.4 |
| Totals | 170.99 | | 1732.5 | 100 |

activity. Fifty-three of these, representing a minimum of six individuals, were identifiable as coming from the extinct giant island deer mouse (*Peromyscus nesodytes*). This mouse species may have been extinct on the Northern Channel Islands by ~8,000 years ago (or possibly later, see [37]), several millennia after a human introduction of a smaller mainland deer mouse, *Peromyscus maniculatus* approximately 11,000 years ago [38]. None of the identifiable rodent bones from the ~11,700 year old component at CA-SRI-26 appear to be from *P. maniculatus*, supporting the integrity of the Paleocoastal assemblage from the A4 soil. All the mouse bones found at an ~8,400 year old village site (CA-SRI-666) on southeastern Santa Rosa Island, in contrast, were from *P. maniculatus* [39].

A rough idea of the relative significance of faunal classes represented in the samples from CA-SRI-26 can be gained by converting the bone and shell weights of the likely prey species (mice and other small mammals excluded) to estimated edible meat yields (Table 3; see [20, 40]). After conversions, birds are estimated to have produced approximately 54 percent of the edible meat represented by the recovered faunal remains, fish 29 percent, marine mammals 12 percent, and shellfish just 5 percent. Thus, birds and fish dominate the meat yields and, along with marine mammals, contributed roughly 95 percent of the edible meat represented in the assemblage.

These estimated dietary yields provide some sense of the importance and diversity of aquatic resources to the site occupants. The relative abundance of various faunal classes may be affected by the Schlepp Effect, however, in which field butchering of certain animals obtained far from the site may result in the under-representation of some taxa. Marine mammal remains would have had to have been transported roughly 4–6 km from the coast, for instance, and probably were butchered on or near the beach, with only prime cuts of meat and blubber transported to the site. Jazwa et al. [41] argued that large red abalone shells were likely to be butchered if collected more than about 3 km from a site, so they may also be under-represented. The identified fish at the site, all common in nearshore marine habitats, are much smaller than pinnipeds and may have been transported as whole or nearly whole carcasses. Finally, the waterfowl were probably captured closer to the site, possibly in wetlands in lower Arlington Canyon, and may be the most likely faunal class to have been carried to the site as whole carcasses.

The heavy representation of birds is consistent with the several crescents found among the bifaces from the site. Bird bones and crescents were also abundant at CA-SRI-512, a Paleocoastal site of comparable age located in a similar blufftop setting ~400 m northeast of CA-SRI-26. The presence of goose bones at both sites suggests that their occupations took place in late fall, winter, or early spring, when migratory geese are known to visit the islands historically [5].

Finally, we collected and processed a small (5.5 liters) sample of soil from the Paleocoastal component at CA-SRI-26 for flotation and archaeobotanical analysis. Only minute amounts of carbonized remains were recovered, including two termite coprolites and one heavily distorted

and unidentified seed. No wood charcoal was recovered, but the carbonized termite coprolites suggest the use of wood as fuel. Similarly, few plant remains were recovered from the buried component at CA-SRI-512, where the paleosol containing Paleocoastal materials also showed evidence of heavy argilliturbation (shrinking and swelling of clay soils from dry to wet cycles). Given the recent identification of carbonized geophyte and other edible plant remains in a ~11,500 year old paleosol at archaeological site CA-SRI-997/H on eastern Santa Rosa Island [16], the dearth of carbonized plant remains in the Paleocoastal components at both CA-SRI-26 and CA-SRI-512 seem likely to be due to a lack of preservation rather than a lack of plant use by the site occupants.

## Summary and conclusions

Our investigations suggest that Paleocoastal peoples camped at CA-SRI-26 roughly 11,700 years ago, occupying a landform several kilometers from the northern shore of Santarosae Island. They chose a high bluff adjacent to Arlington Canyon, one of the largest drainages on the island, where freshwater was readily available along with a commanding view of the coastal lowlands, wetland habitats, and nearshore waters. While living at the site, they hunted birds and marine mammals, fished in kelp forests and other nearshore waters, and collected shellfish from distant rocky shore habitats. Several of the species harvested (e.g., red abalone) are kelp forest obligates, testifying to the importance of kelp forest habitats.

These early maritime peoples were equipped with finely made stemmed points and crescents similar to specimens found in other early NCI Paleocoastal sites, as well as WST sites found near wetland environments across much of western North America [5, 9]. Some scholars have argued that early Paleocoastal people on the NCI may have been mainlanders travelling to the islands seasonally (see [10, 11]), but more than 100 Paleocoastal sites have now been identified on the islands and the distinctive and delicate stemmed CIA and CIB points they used have no parallel in assemblages from adjacent mainland sites (see [20, 42]). Although located on an interior upland landform, the Paleocoastal people who occupied CA-SRI-26 had a maritime economy and hunting technology. Despite the 'maritime' label, however, they were clearly opportunistic, taking advantage of inland locations where they could focus on hunting migratory waterfowl that wintered on Santarosae. Terrestrial plant foods likely were also used, although their remains are poorly preserved in the site soils.

The artifact assemblage from CA-SRI-26 is relatively small, but it is similar to a larger assemblage from nearby CA-SRI-512 that is roughly the same age [5]. Distinct from Paleoindian fluted point technologies in North America, the WST technology of the NCI may have a shared origin with other WST assemblages found in much of western North America and Northeast Asia [5]. Currently, it seems most likely that the island Paleocoastal assemblages may be a specialized maritime variant of the WST. This maritime technology, along with the remains of waterfowl, fish, marine mammals, and shellfish at CA-SRI-26 and other early Paleocoastal sites, suggests that these early maritime peoples were fully adapted to Channel Island ecosystems, probably living on the islands on a year-round basis [9]. Although terrestrial protein sources were limited on the Channel Islands, during the Late Holocene marine fishes, mammals, and birds, along with terrestrial plants (especially geophytes), supported dense hunter-gatherer populations with sophisticated interaction systems, exchange networks, and hierarchical sociopolitical organization [43, 44]. The recovery of a wide variety of marine foods and geophytes in island Paleocoastal assemblages demonstrates that island resources would have easily sustained Paleocoastal peoples throughout the year [5, 9, 16].

Although scholars continue to debate the routes and timing of the initial colonization of the Americas, with evidence for human settlement extending to ~14,000 or perhaps even 16,000

years ago or more (see [4, 6]), a coastal route is now more viable than ever ([1]; see [45, 46] for a debate). Given that NCI Paleocoastal sites postdate the earliest arrival of people in the Americas by a few millennia, CA-SRI-26 and other North American sites dated to ~13,000–11,000 years ago do not confirm the early coastal peopling of the Americas, but they do demonstrate that early maritime adaptations were well established alongside Clovis and Folsom traditions in the American interior [11]. With post-glacial sea level rise posing significant challenges to the preservation and discovery of Late Pleistocene coastal sites, evidence for earlier coastal sites is likely submerged, making the discovery and excavation of sites like CA-SRI-26 vital to our understanding of early coastal peoples in the Americas [1]. Before it is lost to erosion, we hope that further work at the Paleocoastal component at CA-SRI-26 will expand the sample of artifacts and faunal remains recovered, along with our understanding of early Paleocoastal lifeways on California's Channel Islands.

## Acknowledgments

At Channel Islands National Park, logistical and administrative support was provided by Russell Galipeau, Ann Huston, Kelly Minas, and Mark Senning. In the field, we were assisted by Chumash consultants Paula Pugh, Mark Alow Garcia, Quntan Shup, and Gilbert Unzueta, as well as UO graduate students (at the time) Tracy Garcia, Nick Jew, and Lauren Willis. At the UO, Nick Jew helped analyze the chipped stone artifacts, Madonna Moss identified a small sample of fish bones, Molly Casperson and Chelsea Buell identified bird bones, and Beverly Fernandes and Amy Dawson assisted in the lab. All necessary permits were obtained for the described study, which complied with all regulations. Finally, we are grateful to Karen Hardy, Geoff Bailey, Amy Gusick, and an anonymous reviewer for help in the review and revision process.

## Author Contributions

**Conceptualization:** Jon M. Erlandson, Torben C. Rick.

**Data curation:** Jon M. Erlandson.

**Formal analysis:** Jon M. Erlandson, Todd J. Braje, Amira F. Ainis, Brendan J. Culleton, Kristina M. Gill, Courtney A. Hofman, Douglas J. Kennett, Leslie A. Reeder-Myers, Torben C. Rick.

**Funding acquisition:** Jon M. Erlandson, Torben C. Rick.

**Investigation:** Jon M. Erlandson, Todd J. Braje, Torben C. Rick.

**Methodology:** Jon M. Erlandson, Todd J. Braje, Torben C. Rick.

**Project administration:** Jon M. Erlandson, Torben C. Rick.

**Resources:** Jon M. Erlandson, Torben C. Rick.

**Supervision:** Jon M. Erlandson, Torben C. Rick.

**Visualization:** Torben C. Rick.

**Writing – original draft:** Jon M. Erlandson, Todd J. Braje.

**Writing – review & editing:** Jon M. Erlandson, Todd J. Braje, Amira F. Ainis, Brendan J. Culleton, Kristina M. Gill, Courtney A. Hofman, Douglas J. Kennett, Leslie A. Reeder-Myers, Torben C. Rick.

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
