## [Decision Letter · Decision Letter 0]

13 Aug 2020

PONE-D-20-20769

Maritime Paleoindian technology, subsistence, and ecology at an ~11,700 year old Paleocoastal site on California’s Northern Channel Islands, USA

PLOS ONE

Dear Dr. Rick,

Thank you for submitting your manuscript to PLOS ONE. After careful consideration, we feel that it has merit but does not fully meet PLOS ONE’s publication criteria as it currently stands. Therefore, we invite you to submit a revised version of the manuscript that addresses the points raised during the review process.

This is a very interesting paper that was well received by the reviewers.  However, reviewer 3 has made some quite detailed comments and I ask that you address these, before resubmission.

We look forward to receiving your revised manuscript.

Kind regards,

Karen Hardy

Academic Editor

PLOS ONE

Journal Requirements:

2. In your manuscript, please provide additional information regarding the specimens used in your study. Ensure that you have reported specimen numbers and complete repository information, including museum name and geographic location.

For more information on PLOS ONE's requirements for paleontology and archaeology research, see https://journals.plos.org/plosone/s/submission-guidelines#loc-paleontology-and-archaeology-research.

"Our research at CA-SRI-26 was supported by a National Science Foundation grant (BCS 0917677, to Erlandson and Rick), our home institutions, Channel Islands National Park (CINP), and funds from the Paleoindian Research Endowment at the University of Oregon’s (UO) Museum of Natural & Cultural History."

"1) JME, TCR, National Science Foundation, BCS-0917677, NO

2) JME, US National Park Service, Cooperative Agreement H812005033, NO"

4. We note that Figure 1 in your submission contain map images which may be copyrighted. All PLOS content is published under the Creative Commons Attribution License (CC BY 4.0), which means that the manuscript, images, and Supporting Information files will be freely available online, and any third party is permitted to access, download, copy, distribute, and use these materials in any way, even commercially, with proper attribution. For these reasons, we cannot publish previously copyrighted maps or satellite images created using proprietary data, such as Google software (Google Maps, Street View, and Earth). For more information, see our copyright guidelines: http://journals.plos.org/plosone/s/licenses-and-copyright.

4.1.    You may seek permission from the original copyright holder of Figure 1 to publish the content specifically under the CC BY 4.0 license. 

4.2.    If you are unable to obtain permission from the original copyright holder to publish these figures under the CC BY 4.0 license or if the copyright holder’s requirements are incompatible with the CC BY 4.0 license, please either i) remove the figure or ii) supply a replacement figure that complies with the CC BY 4.0 license. Please check copyright information on all replacement figures and update the figure caption with source information. If applicable, please specify in the figure caption text when a figure is similar but not identical to the original image and is therefore for illustrative purposes only.

5. Please include your tables as part of your main manuscript and remove the individual files. Please note that supplementary tables (should remain/ be uploaded) as separate "supporting information" files.

Reviewers' comments:

Reviewer's Responses to Questions

**Comments to the Author**

1. Is the manuscript technically sound, and do the data support the conclusions?

Reviewer #1: Yes

Reviewer #2: Yes

Reviewer #3: Yes

2. Has the statistical analysis been performed appropriately and rigorously? 

Reviewer #1: Yes

Reviewer #2: Yes

Reviewer #3: N/A

3. Have the authors made all data underlying the findings in their manuscript fully available?

Reviewer #1: Yes

Reviewer #2: Yes

Reviewer #3: Yes

4. Is the manuscript presented in an intelligible fashion and written in standard English?

Reviewer #1: Yes

Reviewer #2: Yes

Reviewer #3: Yes

5. Review Comments to the Author

Reviewer #1: Erlandson and his colleagues present data from one of the pre-11,000 BP sites on the northern Channel Islands and its analysis with respect to technology, subsistence, and environment. Their paper expands the available information concerning the early Paleocoastal people who occupied the large channel island that existed at that time. The data and analysis are clearly presented, and I can offer no suggestions for their improvement. I noticed, however, that the LcLaren et al. reference cited on line 67 is missing from the References.

Reviewer #2: This manuscript is an excellent addition to the literature on early human occupation along the Pacific Coast of the Americas. Over the past decade, evidence for a terminal Pleistocene pacific coastal migration and subsequent occupation has grown. Yet much of the evidence comes from relatively small sites that none-the-less suggest an established occupation of groups focused on a variety of opportunistic marine and wetland habitats. The site detailed in this manuscript, SRI-26, was excavated by the authors on Santa Rosa Island and provides additional evidence from which to characterize the early occupations found on a hot spot for terminal Pleistocene and Early Holocene occupation in the New World, the Northern Channel Islands of California.

Although small, the site assemblage provides critical data that informs on seasonal movements and habitat use for early occupants that show adept use of a variety of habitats - some of which are no longer extant due to sea level rise - and species that have since become extinct or are threatened today. While the faunal data from this work do raise questions on accurate representations of Paleocoastal diet within these early sites due to issues of erosion and the Schlepp Effect, the technological assemblage, specifically chipped stone crescents, from SRI-26 again provide a link between island technologies and the Western Pluvial Lakes tradition identified on the mainland. Although the technological assemblages have similarities, the CIA and CIB points from SRI-26 are characteristic of those found in other terminal Pleistocene sites on the island and remain unique to island assemblages. This further discourages the argument that the island assemblages were made by mainland groups, conducting forays to the island for resources.

The data presented support the conclusions. Important for reporting of terminal Pleistocene-aged sites is veracity of the dating methods from secure contexts. The authors have complementary dates from both AMS 14C dates and collagen dates placing one of the site loci within the terminal Pleistocene. Other dates obtained from additional loci are within the early Holocene, but the tables indicate that the faunal data are from the terminal Pleistocene-dated locus. The CIA point identified, though just outside of the intact site deposit is an excellent example of this technology that is considered evidence of terminal Pleistocene occupation. Analysis of the faunal material is reported as are the meat weight calculation used for relative edible meat yield for the shell and bone identified in the site. Again, the authors do note that these totals may be impacted by preservation and representational issues, but what does remains of the food remains suggest birds are an important food source, a trend identified in other terminal Pleistocene sites, but not in island sites dating later in time. Unfortunately, archaeobotanical work at the site did not yield material to discuss exploitation of plants during site occupation, but this is not surprising for these early sites due to preservation issues.

The conclusions presented sum up the findings and suggest that the while the early peoples at this site were considered maritime, the site would have been about 4-6 km back from the coast at the time of occupation. This positioning may have been for targeting migratory waterfowl, and perhaps collecting plant resources, while making trips to the coast and the abundant resources there. This interpretation further supports the notion that preservation of other faunal remains such as shellfish, fish, and sea mammal, which are less abundant than birds, may have been impacted by butchering near to the coast and carrying only meat back to the site. This research provides an excellent addition to the growing body of data showing that island Paleocoastal assemblages are distinct from interior Paleoindian assemblages, and likely date to even earlier in time. The data presented also help to inform on the mobility and subsistence pursuits of the early mobile hunter-gatherer groups present along the Pacific coast and offshore islands during the terminal Pleistocene. This further clarifies which landscape features and regions where additional terminal Pleistocene sites may be identified and sampled, before losing more material to erosional processes.

Reviewer #3: This is a clear and well-supported presentation of results from the archaeological site of CA-SRI-26, an 11.7 ka site on the California Channel Islands with a maritime signature, preserved under a covering of alluvium and exposed to recent investigation by erosion. The site setting, stratigraphy, dating, methods, and evidence of technology and subsistence and their similarities and differences in a wider comparative setting are all set out clearly and to a high standard with all the information one would expect.

The significance of the site is that it is of relatively early date for a site with evidence for use of marine resources and advances knowledge of subsistence and settlement patterns in the Channel Islands for this period in comparison with other known sites, where only stone tools or human bone are present. I think this site is the earliest site anywhere on the Pacific coast (not only on the Channel Islands) with actual archaeological evidence of marine food remains. If that is so, then this point should receive greater emphasis.

There are some minor omissions and errors or inconsistencies, which I set out below.

There are two issues that need to be addressed with some further discussion and detail, and both have to do with the wider significance of this site in a world context and for an international audience who are not necessarily familiar with all the details of the American scene.

The first issue is the nature and duration of settlement at CA-SRI-26. Is it a year-round settlement or a seasonal occupation? The Abstract refers to a 'relatively brief occupation' in winter (Lines 51-52), the text refers to the site as evidence that the people who used it lived on the Islands on a 'year-round basis' (line 341). These seem at least partially contradictory. How good is the evidence for year-round occupation, either at CA-SRI-26 or on the islands generally? Given that there are no land mammals (or none that were hunted) on the islands, and little plant food, at least given what is preserved, is there enough food on the islands to support a decent-size population all year round? It seems that marine resources would have to do the job. Yet the remains at CA-SRI-26 are quite meager. This may be due, as the authors say, to schlepp effects, the distance of the site, 4-6 km, from the sea coast, and the likelihood that processing sites with more abundant marine-resource remains are on the seacoast. If that is so, then such sites would presumably be now well below present sea level. That point might be worth making, along with some assessment of the prospects that such sites might have survived (for example beneath marine sediment) and be discoverable. Are there precedents from later periods or ethnography for island populations capable of sustained self-support based mainly on marine resources? Would food storage be required? Would such populations exist in isolation from populations on the mainland? Otherwise the case for year-round settlement seems to rest on the differences in the lithic assemblages compared to the mainland. There is something here that needs some (brief) elaboration.

This raises the second issue, and perhaps the main one. What is the significance of this site in relation to the wider debates about the coastal colonization hypothesis for the Americas? The authors raise this briefly in the Introduction, and in the summary their main point is that the new evidence refutes the ideas of Balter and Yesner that the island sites are just seasonal visits by mainlanders. Balter is a science journalist and Yesner has long been skeptical that maritime lifeways existed before the mid-Holocene, so I'm not sure that their views are that significant anyway. In any case, seasonal visits to the islands would not necessarily refute the hypothesis of earliest coastal colonization (if that is what Balter and Yesner are claiming). Conversely, year-round maritime occupation of the islands does not seem sufficient to necessarily support the coastal colonization hypothesis (the Islands might have been occupied at a later date than the earliest wave of colonization). The main problem here is the dates. If the NCI are on the kelp highway, as referred to in the text, why are there no sites earlier than about 13 ka, given that we have dates back to about 14.5 to 15 ka elsewhere in the Americas (Cooper's Creek, Monte Verde, etc., which, incidentally. are inland and not marine-based). Are the authors over-interpreting the site to make it seem more significant than is actually the case? Or is there more to be said to put the site findings into the wider context of earliest colonization? I think some further discussion, together with reference to more of the wider literature (Potter et al. comes to mind), is needed

Minor Details:

The following references are missing: Peltier 2015, Argus et al. 2014, Davis et al. 2019, McClaren et al. 2019

Line 207. the Far West. What region does this refer to?

Lines 209-210. '...assemblage is smaller than CA-SRI-512 but similar...to the larger collection from CA-SRI-512'. Something wrong here with site numbers

Line 239 'greater faunal diversity', greater than what?

Lines 278-279 (and Table 3). birds are 54% and fish 29%, but in line 281 text it is said that birds and fish contributed 95%. Percentage in text looks wrong?

6. PLOS authors have the option to publish the peer review history of their article (what does this mean?). If published, this will include your full peer review and any attached files.

Reviewer #1: No

Reviewer #2: **Yes: **Amy E Gusick

Reviewer #3: No

---

## [Author Response · Author response to Decision Letter 0]

19 Aug 2020

I uploaded a separate response to reviewers file with clear point by point assessment of the comments.

---

## [Decision Letter · Decision Letter 1]

21 Aug 2020

PONE-D-20-20769R1

Maritime Paleoindian technology, subsistence, and ecology at an ~11,700 year old Paleocoastal site on California’s Northern Channel Islands, USA

PLOS ONE

Dear Dr. Rick,

Thank you for submitting your manuscript to PLOS ONE. After careful consideration, we feel that it has merit but does not fully meet PLOS ONE’s publication criteria as it currently stands. Therefore, we invite you to submit a revised version of the manuscript that addresses the points raised during the review process.

The reviewer has added one comment, or rather, clarification,  that he asks to be addressed. 

We look forward to receiving your revised manuscript.

Kind regards,

Karen Hardy

Academic Editor

PLOS ONE

Reviewers' comments:

Reviewer's Responses to Questions

**Comments to the Author**

1. If the authors have adequately addressed your comments raised in a previous round of review and you feel that this manuscript is now acceptable for publication, you may indicate that here to bypass the “Comments to the Author” section, enter your conflict of interest statement in the “Confidential to Editor” section, and submit your "Accept" recommendation.

Reviewer #3: (No Response)

2. Is the manuscript technically sound, and do the data support the conclusions?

Reviewer #3: Yes

3. Has the statistical analysis been performed appropriately and rigorously? 

Reviewer #3: Yes

4. Have the authors made all data underlying the findings in their manuscript fully available?

Reviewer #3: Yes

5. Is the manuscript presented in an intelligible fashion and written in standard English?

Reviewer #3: Yes

6. Review Comments to the Author

Reviewer #3: The authors have satisfactorily addressed all my comments except one, and that is a minor detail, but one that is potentially confusing

I refer to lines 219-220: "The CA-SRI-26 assemblage is smaller than those from CA-SRI-512 and Cardwell Bluffs, but similar

in many respects to the [larger] collection from nearby CA-SRI-512."

My original query about this sentence was not about the use of ‘larger’ but about the fact that the sentence seemed to be comparing two assemblages in such a way as to suggest that they are from two different sites, yet both have the same site label CA-SRI-512. It’s the use of the label CA-SRI-512 that I was querying. Were the authors referring to two distinct assemblages from the same site? Or two different sites? And if two different sites, why do they have the same site record number?

On re-reading the sentence in context with the preceding paragraph, I see where the confusion lies. It's the use of CA-SRI-512 twice in the same sentence. There are three sites: CA-SRI-26, CA_SRI-512, which has a larger assemblage and is nearby, and Cardwell Bluffs (CA-SMI-678 et seq.), which is further away. I think the following wording would avoid the potential confusion: "The CA-SRI-26 assemblage is similar in many respects to the nearby assemblage from CA-SRI-512, though smaller, and similar to the Cardwell Bluffs assemblage."

I now see how I misunderstood the authors' original sentence, and how they in turn have misunderstood my comment and in particular my use of 'numbers', which I used to refer to CA site labels, and they, I think, interpreted as referring to quantities of material. Truly are we divided by a common language!

7. PLOS authors have the option to publish the peer review history of their article (what does this mean?). If published, this will include your full peer review and any attached files.

Reviewer #3: **Yes: **Geoff Bailey

---

## [Author Response · Author response to Decision Letter 1]

21 Aug 2020

We have edited the requested sentence to:

"The CA-SRI-26 assemblage is smaller, but similar in many respects to the assemblages from CA-SRI-512 and, to an extent, Cardwell Bluffs."

---

## [Editor Report · Decision Letter 2]

26 Aug 2020

Maritime Paleoindian technology, subsistence, and ecology at an ~11,700 year old Paleocoastal site on California’s Northern Channel Islands, USA

PONE-D-20-20769R2

Dear Dr. Rick,

We’re pleased to inform you that your manuscript has been judged scientifically suitable for publication and will be formally accepted for publication once it meets all outstanding technical requirements.

Kind regards,

Karen Hardy

Academic Editor

PLOS ONE
---

## [Editor Report · Acceptance letter]

1 Sep 2020

PONE-D-20-20769R2 

Maritime Paleoindian technology, subsistence, and ecology at an ~11,700 year old Paleocoastal site on California’s Northern Channel Islands, USA 

Dear Dr. Rick:

I'm pleased to inform you that your manuscript has been deemed suitable for publication in PLOS ONE. Congratulations! Your manuscript is now with our production department. 

Kind regards, 

on behalf of

Dr. Karen Hardy 

Academic Editor

PLOS ONE